# The Role of the Pathogen Dose and PI3Kγ in Immunometabolic Reprogramming of Microglia for Innate Immune Memory

**DOI:** 10.3390/ijms22052578

**Published:** 2021-03-04

**Authors:** Trim Lajqi, Christian Marx, Hannes Hudalla, Fabienne Haas, Silke Große, Zhao-Qi Wang, Regine Heller, Michael Bauer, Reinhard Wetzker, Reinhard Bauer

**Affiliations:** 1Institute of Molecular Cell Biology, Jena University Hospital, D-07745 Jena, Germany; trim.lajqi@med.uni-heidelberg.de (T.L.); fabienne.haas@med.uni-jena.de (F.H.); grosse_silke@gmx.de (S.G.); regine.heller@med.uni-jena.de (R.H.); 2Department of Neonatology, Heidelberg University Children’s Hospital, D-69120 Heidelberg, Germany; hannes.hudalla@med.uni-heidelberg.de; 3Leibniz Institute on Ageing, Fritz Lipmann Institute, D-07745 Jena, Germany; christian.marx@leibniz-fli.de (C.M.); zhao-qi.wang@leibniz-fli.de (Z.-Q.W.); 4Department of Anesthesiology and Intensive Care Medicine, Jena University Hospital, D-07747 Jena, Germany; michael.bauer@med.uni-jena.de (M.B.); reinhard.wetzker@uni-jena.de (R.W.)

**Keywords:** microglia, immunometabolism, innate immune memory, LPS, PI3Kγ, glycolysis, pentose phosphate pathway, OXPHOS, OCR, ECAR

## Abstract

Microglia, the innate immune cells of the CNS, exhibit long-term response changes indicative of innate immune memory (IIM). Our previous studies revealed IIM patterns of microglia with opposing immune phenotypes: trained immunity after a low dose and immune tolerance after a high dose challenge with pathogen-associated molecular patterns (PAMP). Compelling evidence shows that innate immune cells adopt features of IIM via immunometabolic control. However, immunometabolic reprogramming involved in the regulation of IIM in microglia has not been fully addressed. Here, we evaluated the impact of dose-dependent microglial priming with ultra-low (ULP, 1 fg/mL) and high (HP, 100 ng/mL) lipopolysaccharide (LPS) doses on immunometabolic rewiring. Furthermore, we addressed the role of PI3Kγ on immunometabolic control using naïve primary microglia derived from newborn wild-type mice, PI3Kγ-deficient mice and mice carrying a targeted mutation causing loss of lipid kinase activity. We found that ULP-induced IIM triggered an enhancement of oxygen consumption and ATP production. In contrast, HP was followed by suppressed oxygen consumption and glycolytic activity indicative of immune tolerance. PI3Kγ inhibited glycolysis due to modulation of cAMP-dependent pathways. However, no impact of specific PI3Kγ signaling on immunometabolic rewiring due to dose-dependent LPS priming was detected. In conclusion, immunometabolic reprogramming of microglia is involved in IIM in a dose-dependent manner via the glycolytic pathway, oxygen consumption and ATP production: ULP (ultra-low-dose priming) increases it, while HP reduces it.

## 1. Introduction

Microglia, the self-renewing innate immune cells of the central nervous system (CNS), exhibit immune memory with implications for brain homeostasis and pathologies. This resident macrophage-like population confers innate immunity to the CNS as the first line of defense against invading pathogens [1,2]. In addition, they control the neuronal patterning and wiring of the brain in early development and fulfill supportive functions for maintenance of tissue homeostasis, tissue support, neuroplasticity and neuroprotection due to their ability to surveil the microenvironment for alterations [3,4,5,6,7].

Recently, peripheral inflammatory insults in rodents have been shown to induce long-term changes in microglial responsiveness as a feature of innate immune memory (IIM) [8]. Several seminal reports revealed memory-like immune responses of microglia toward pathogen-associated molecular patterns (PAMPs). For example, Schaafsma et al. discovered immune tolerance after microglial priming with LPS. Mechanistic analysis revealed epigenetic modifications as essential mediators of the immune-suppressed phenotype [9]. Wendeln et al. reported that peripherally applied inflammatory stimuli induce acute immune training and tolerance in the brain and lead to differential epigenetic reprogramming of microglia [8]. Our previous studies underlined these findings and revealed dose-dependent immune responses of microglia with opposing immune phenotypes, e.g., trained immunity after initial contact with low doses of PAMP and immune tolerance after high PAMP doses [10]. Compelling evidence shows that epigenetic reprogramming provides a molecular basis for immune memory in microglia [8,9,10]. Therefore, the capacity of microglia to develop IIM, that is, longer lasting molecular reprogramming of responsiveness to immune stimuli, has been established [1].

Accumulating evidence shows that innate immune cells adopt long-term inflammatory phenotypes after brief encounters with pathogen- or danger-associated stimuli as a result of immunometabolic switches [11,12,13]. IIM induction upon stimulation relies on an active interplay between epigenetic and metabolic reprogramming of the innate immune cells. This may involve reprogramming of hematopoietic stem and progenitor cells in the bone marrow [14]. Furthermore, an upregulation of glycolysis in immune-competent cells upon proinflammatory activation has been observed [15,16,17]. This may trigger epigenetic reprogramming via histone modifications such as H3K4me3 leading to enhanced transcription of proinflammatory genes. In addition, the increased ratio of oxidized to reduced nicotinamide adenine dinucleotide phosphate (NAD^+^/NADH ratio) due to enhanced glycolysis affects epigenetic regulation via the action of NAD^+^-dependent histone deacetylases/sirtuins [13]. However, while evidence for epigenetic reprogramming has been substantiated as a mechanism underlying IIM in microglia [8,9], the contribution of immunometabolic reprogramming is less clear. 

Phosphoinositide 3-kinase γ (PI3Kγ) is a member of class I phosphoinositide 3-kinases (PI3Ks). PI3Ks are classified into two subclasses: class IA, which has three members, PI3Kα, β and δ; and class IB, which has one member, PI3Kγ. PI3Kα and δ isoforms are activated by receptor tyrosine kinases, while PI3Kγ is activated by G protein β–γ subunits (Gβγ), which are usually derived from G_i_-coupled receptors. In contrast, the PI3Kβ isoform is regulated by receptor tyrosine kinases and by Gβγ subunits [18,19,20]. PI3Ks were originally characterized as a signaling protein, which generates phosphatidylinositol 3,4,5-trisphosphate for subsequent protein kinase B/Akt activation [21,22,23]. Ubiquitously expressed PI3Kα and PI3Kβ are vital for cell growth, division and survival [24,25] and genetic ablations of each resulted in early embryonic lethality [26,27]. PI3Kδ, also mainly expressed in hematopoietic cells, cooperates with PI3Kγ in immune-mediated inflammation by a complex epistatic interaction with activities critical during the process of inflammatory diseases [28,29,30].

PI3Kγ was identified as the major PI3K catalytic isoform in primary myeloid cells, as these cells express at least 25-fold more PI3Kγ than other isoforms [31]. Studies in PI3Kγ-deficient mice revealed impaired respiratory burst and motility of peripheral leukocytes indicating a major regulatory function of PI3Kγ in immune cells [32]. Others and we revealed robust expression of PI3Kγ in primary microglial cells [33,34]. We found specific impact of PI3Kγ on microglial migration, control of matrix metalloproteinases and phagocytic activity [34,35,36,37]. Furthermore, our studies demonstrated an intimate interplay of PI3Kγ with cAMP signaling pathways via PI3Kγ-mediated activation of phosphodiesterases (PDEs) leading to inhibition of inflammatory responses [34,35,36,38,39]. 

The present report extends our previous studies on dose-dependent induction of IIM in microglia by evaluating the role of immunometabolism in the microglial IIM. Furthermore, we examined the specific role of lipid kinase-dependent and -independent functions of PI3Kγ for the metabolic control of IIM. The objective of this manuscript was to illustrate the quite complex patterns of metabolic reprogramming in microglia for innate immune memory depending on the priming dose and PI3Kγ signaling. This includes as the main components the glycolytic metabolic pathway, the tricarboxylic acid (TCA) cycle and the electron transport chain, the pentose phosphate pathway and fatty acid oxidation, which we addressed accordingly. Primary microglial cells from neonatal wild-type and PI3Kγ knockout mice (PI3Kγ^-/-^) as well as from mice carrying a targeted mutation leading to the loss of lipid kinase activity (PI3Kγ^KD/KD^) were employed. Microglial cells received LPS stimulation after six days of LPS priming with ultra-low and high doses (ULP or HP, respectively) and were analyzed for immunometabolic rewiring in comparison to cells without priming. Our data reveal that LPS-induced IIM induced by ULP was accompanied by a marked enhancement of oxygen consumption rate (OCR) and increased ATP production. In contrast, HP priming led to suppressed OCR and glycolytic activity indicative of immune tolerance. Intriguingly, the signaling protein PI3Kγ provoked a marked inhibition of glycolysis possibly due to modulation of cAMP-dependent pathways. However, immunometabolic rewiring due to dose-dependent LPS priming was not related to specific PI3Kγ signaling events.

## 2. Results

### 2.1. Energy Metabolism in LPS-Induced IIM

Metabolic profiling using a Seahorse analyzer revealed that ULP induced a marked enhancement of OCR and an increased ATP production in naïve microglial cells derived from newborn C57BL/6 mice (Figure 1B,D). Intriguingly, ULP provoked a maximal OCR stimulation which was not further enhanced by uncoupling the respiratory chain activity from ATP synthesis using 2,4-dinitrophenol (DNP). Accordingly, a clear increase of respiration in ULP microglia was seen when compared to unprimed microglia (Figure 1G).

Moreover, ULP induced a marked stimulation of glycolytic reserve capacity indicating the capability of microglia to respond to an additional energy demand (Figure 1H–J). In contrast, HP induced a suppression of OCR and reduction of ATP production, maximal respiration and non-mitochondrial respiration. Similarly, the glycolytic reserve capacity was suppressed by HP as revealed by reduced extracellular acidification rates (ECAR) during maximal respiration (Figure 1J).

### 2.2. Aerobic Glycolysis and β-Oxidation in LPS-Induced IIM

The opposing priming effects of ULP and HP on microglial ECAR responses as characterized by the Seahorse analysis were confirmed by expression patterns of key (partly rate-controlling) glycolytic enzymes (Figure 2A–E). 

ULP was accompanied by increased gene expression of enzymes throughout the glycolytic pathway (aldolase A, enolase 2, hexokinase 2, phosphofructokinase 1 (PFK1) and pyruvate kinase M2 (PKM2)), whereas HP led to a downregulation of the respective genes. The altered gene expression patterns induced by ULP and HP may explain the observed changes in the ECAR. In line with this, an increased lactate production due to ULP and a reduction of lactate production after HP were measured (Figure 2F). To elucidate whether LPS-induced microglial priming affects long-chain fatty acid oxidation (LC-FAO), the oxidation of radioactively labeled palmitate was assessed (Figure 2G). ULP led to a reduction of the LC-FAO rate, whereas HP induced a considerable increase. Hence, ULP and HP triggered opposing effects on cellular energy metabolism with alterations indicative of a proinflammatory phenotype after ULP and of an anti-inflammatory phenotype after HP.

### 2.3. Role of PI3Kγ on Priming-Dependent Metabolic Rewiring in Microglial Cells

In order to elucidate a possible role of PI3Kγ in priming effects of different LPS doses on energy metabolism, microglia isolated from wild-type mice were compared with microglia from PI3Kγ^-/-^ and PI3Kγ^KD/KD^ mice. First, we studied gene expression of the major enzymes which are involved in glycolysis (PFK1, PKM2 [40,41]; hypoxia-inducible factor 1 alpha (HIF-1α) [42]), the pentose phosphate pathway (PPP) (carbohydrate kinase-like protein (CARKL) [43]), the tricarboxylic acid (TCA) cycle (isocitrate lyase 1 (ICL1) [44]) and oxidative phosphorylation (OXPHOS) (NADH dehydrogenase 1 alpha subcomplex subunit 9 (NDUFA9), succinate dehydrogenase complex, subunit A (SDHA) [45]).

As shown in Figure 3A–C, PI3Kγ is involved in the regulation of metabolic enzyme expression. Complete depletion of PI3Kγ led to a markedly increased gene expression of both rate-controlling glycolytic enzymes PFK1 and PKM2 under resting conditions, which was further enhanced by LPS stimulation (UP state) when compared to wild-type microglia. In contrast, the expression of these enzymes was not different between PI3Kγ^KD/KD^ microglia and wild-type cells. Similar observations were made for HIF-1α. The effect of PI3Kγ knockout on glycolytic enzyme expression correlated with an increase of cAMP (see Appendix A) in these cells suggesting an involvement of cAMP-dependent pathways. Of note, HP reduced the LPS-induced increase of glycolytic enzyme expression when compared to unprimed cells suggesting induction of immune tolerance by HP. ULP did not alter gene expression of the glycolytic enzymes in PI3Kγ-deficient or PI3Kγ^KD/KD^ microglia when compared to unprimed cells.

CARKL, an enzyme of the PPP, was upregulated by ULP and downregulated by HP in wild-type and PI3Kγ^KD/KD^ microglia (Figure 3D). In unprimed PI3Kγ-deficient cells, LPS induced a strong upregulation of CARKL, which was reduced after ULP or HP priming. Gene expression of ICL1, an enzyme of the TCA cycle, was enhanced by LPS stimulation, more markedly in PI3Kγ-deficient cells, but was little affected by priming (Figure 3E). The expression of NDUFA9, a component of the electron transport chain (ETC) (Figure 3F), showed comparable regulation as ICL1 by LPS, PI3Kγ depletion and priming. However, SDHA was downregulated in the absence of PI3Kγ or in the presence of PI3Kγ^KD/KD^. Under these conditions, ULP-induced upregulation, as well as HP-induced inhibition, occurred in wild-type microglia (Figure 3G). 

Subsequently, PI3Kγ genotype-related effects on oxygen consumption and glycolytic flux were analyzed. As shown in Figure 4A–H, the OCR in wild-type cells was higher in response to LPS in unprimed cells (UP), further enhanced by ULP and reduced after HP. Microglia derived from PI3Kγ^-/-^ mice showed maximum OCR already under resting conditions without further elevation after LPS stimulation. ULP in PI3Kγ^-/-^ cells was accompanied by a mild but significant OCR reduction compared to the one in wild-type microglia, while HP induced a moderate OCR inhibition, which was less pronounced than the effect in wild-type cells. PI3Kγ^KD/KD^ microglia displayed comparably low OCR values under resting conditions as seen in wild-type microglia and showed only little response after addition of pharmacological compounds interfering with ATP synthesis or ETC (Figure 4A–H). 

Mitochondrial (m) ATP production in wild-type microglia as calculated from OCR values under resting conditions and after addition of oligomycin was low in unstimulated cells, increased after LPS stimulation in unprimed cells and peaked in response to ULP, whereas HP induced a marked reduction (Figure 4I). In contrast, PI3Kγ^-/-^ microglia displayed high mATP production under unstimulated conditions, which was not further increased by LPS stimulation or ULP but showed a significant reduction after HP. The mATP production in PI3Kγ^KD/KD^ microglia under basal conditions was similar to the one in wild-type microglia and not affected by LPS or ULP, while HP induced a significant reduction. Similar differences as described for ATP production were observed for non-mitochondrial respiration, proton leak and maximal respiration between the different genotypes and priming conditions (Figure 4J–L).

To clarify a possible impact of PI3Kγ on glycolysis, we measured ECAR again. In wild-type and PI3Kγ^KD/KD^ microglia, the ECAR was almost unchanged by compounds interfering with ATP synthesis or ETC pointing to low glycolytic capacity under resting conditions (Figure 5A). In contrast, the ECAR in resting or LPS-stimulated PI3Kγ^-/-^ microglia was increased in response to metabolic challenges indicating high glycolytic capacity. 

Intriguingly, ULP, but not HP, increased the glycolytic reserve capacity in wild-type microglia similar to what was measured in PI3Kγ^-/-^ microglia (Figure 5A–I). PI3Kγ^KD/KD^ microglia exhibited minimal differences in the ECAR in response to LPS alone or in combination with ULP. HP provoked a significantly reduced ECAR. 

## 3. Discussion

Microglial cells are “surveillance” cells, in charge of maintaining and safeguarding brain homeostasis. Therefore, their response to an initial PAMP confrontation is highly relevant in order to maintain brain tissue protection [1,2]. Recent studies by our group revealed that microglia from neonatal brains display features of trained immunity and immune tolerance after repeated contact with pathogens in a dose-dependent manner [10,46]. Intriguingly, naïve microglia exhibit a remarkable sensitivity to bacterial PAMPs [47], possibly to prevent hazardous events in newborns, who are highly susceptible to infections and mainly rely on innate immunity [48]. Our findings confirmed the high susceptibility of microglia to PAMPs by showing that priming with ultralow LPS dosages induced elevated proinflammatory responses [10]. In line with this, previous studies showed that LPS is extraordinarily potent and able to stimulate host responses even in the femtomolar range corresponding to about 100 invading gram-negative bacteria [49,50,51]. Therefore, as few as two dozen LPS MD-2/TLR4 complexes, the decisive intracellular signal transfer units, can trigger measurable proinflammatory responses in an individual immune cell, implying very efficient oligomerization of these ternary complexes [52]. 

Epigenetic reprogramming obviously participates in dose-dependent induction of microglial IIM. Previous findings made by us and others suggested that distinct epigenetic histone modifications contribute to microglial IIM in response to ULP and HP stimulation leading to immune training and immune tolerance, respectively [8,9,10,53]. Herein, we report for the first time that microglial immunometabolic rewiring characterized by an enhanced OCR and ATP production as well as maximal respiration and glycolytic reserve capacity is involved in ULP-induced immune training. This was accompanied by a coordinated elevation of gene expression of key enzymes of glycolysis, including rate-controlling enzymes. In contrast, HP led to a reduction of the OCR, ATP production and maximal respiration, which was associated with a distinct downregulation of glycolytic enzymes. Therefore, immune training appears to be metabolically linked to a proinflammatory phenotype [16]. Nevertheless, we have to consider that the mechanistic approach to elucidate the pattern of metabolic reprogramming was focused on early gene regulatory responses of IIM effects. Future studies will examine the importance of such metabolic reprogramming for microglial function.

The involvement of PI3Kγ in dose-dependent trained immunity and immune tolerance was previously elaborated in the same way as herein, e.g., by comparative analysis of microglia isolated from wild-type mice, mice deficient in PI3Kγ expression (PI3Kγ^-/-^) and mice expressing a catalytically inactive mutant of PI3Kγ (PI3Kγ^KD/KD^) [10]. In this study, we revealed that the lipid kinase function of PI3Kγ triggered a biphasic effect of priming in wild-type microglia with increased proinflammatory cytokine release due to ULP and inhibition of cytokine release after HP. These effects were mediated by the protein kinase B/Akt pathway response including NFκB [10]. The present results validate our previous findings of a pronounced impact of PI3Kγ on dose-dependent induction of microglial IIM. However, the immunometabolic effects are mainly associated with the lipid kinase-independent function of PI3Kγ. The reported effects may be related to PI3Kγ-dependent attenuation of the cAMP/protein kinase A pathway via activation of phosphodiesterases, which hydrolyze cAMP to 5′-AMP [34,35,39]. Hence, microglia obtained from PI3Kγ^-/-^ mice displayed enhanced cellular cAMP content regardless of preprocessing suggesting that cAMP-triggered pathways were enhanced (Appendix A).

We performed a targeted transcriptional gene expression analysis to elucidate the PI3Kγ-dependent impact on opposing dose-dependent IIM effects on immunometabolic rewiring. Our data revealed a sophisticated metabolically orchestrated response pattern reflecting the regulation of metabolic pathways that allow coupling of cellular functions to the metabolic machinery [13,16]. Specifically, ULP induced enhanced aerobic glycolysis in wild-type microglia as indicated by increased lactate production and upregulation of key enzymes of this pathway. The latter was mediated by proinflammatory signaling as indicated by corresponding HIF-1α upregulation [42]. Remarkably, despite increased glycolysis, the pentose phosphate pathway (PPP) appears to be suppressed due to enhanced CARKL expression [43,54]. Therefore, generation of riboses for nucleotides, amino acids and nicotinamide adenine dinucleotide phosphate (NADPH) may be inhibited with the latter leading to an altered redox balance [43]. In contrast, OXPHOS is upregulated in response to ULP of wild-type microglia, which may be due to enhanced expression of compounds of the TCA cycle and the ETC. Induction of immune tolerance by HP provoked an opposite response of immunometabolic alterations in wild-type microglia, i.e., a reduced expression of rate-limiting glycolytic enzymes, diminished lactate production and OXPHOS suppression. These findings suggest that immune tolerance is accompanied by energy-saving processes as an adaptive stress response thus supporting cell vitality by reducing energy consumption [55].

Loss of PI3Kγ in microglia is accompanied by a marked increase of energy-demanding processes including proinflammatory activities, possibly driven by a distinct ETC uncoupling. We found that PI3Kγ knockdown led to enhanced glycolytic enzyme expression, increase of the ECAR, HIF-1α upregulation, PPP reinforcement via diminished CARKL expression and TCA inhibition verified by ICL1 downregulation [13,16] under resting conditions. However, in contrast to wild-type microglial cells, in which LPS induced enhanced gene expression of glycolytic enzymes, of the PPP control factor and of compounds of OXPHOS, immunometabolism of PI3Kγ^-/-^ microglia appears to be less dependent on inflammatory stimulation or microglial priming. We found no differences in OCR parameters or gene expression patterns between unprimed and primed LPS-stimulated PI3Kγ^-/-^ microglia. In contrast, microglia with only loss of PI3Kγ’s lipid kinase activity exhibited a different immunometabolic phenotype. While similar immunometabolic features of PI3Kγ^KD/KD^ and wild-type microglia were seen under resting conditions, LPS stimulation alone or in combination with preceding priming failed to alter the immunometabolic response. Neither OCR parameters nor gene expression of enzymes involved in carbohydrate metabolism were changed. Taking these results together, our study clearly underlines the marked role of PI3Kγ in microglial immune control as reported previously [10,34,35,36,37,46], even if mechanistic explanations of the distinct cellular phenotypes are beyond the scope of this study. 

The activation of cAMP/CREB signaling caused by missing PI3Kγ’s scaffold function and hence loss of phosphodiesterase (PDE) activation [39,56] may be responsible for distinct ETC uncoupling. Our immunometabolic data substantiate previous findings that loss of PI3Kγ’s scaffold function is causally related to proinflammatory sequels in different disease models including septic encephalopathy and its hypothermic aggravation [35,57], focal brain ischemia [36] and inflammation-induced myocardial depression in septic shock [38]. However, a molecular link between increased intracellular cAMP levels as well as a related signaling and mitochondrial disturbance with ETC uncoupling remains unknown and cannot be deduced from the present data. 

## 4. Materials and Methods 

### 4.1. Animals and Microglia Isolation Procedures

PI3Kγ knockout mice (PI3Kγ^-/-^) [32] and mice carrying a targeted mutation causing loss of lipid kinase activity (PI3Kγ^KD/KD^) [39] were on the C57BL/6J background for >10 generations. Age-matched C57BL/6 mice were used as controls (wild-type). Newborn (P0–P3) mice from a locally inbred C57Bl/6J mouse strain (n = 261), PI3Kγ^-/-^ (n = 210) and PI3Kγ^KD/KD^ (n = 210) mice were used to isolate primary microglial cells. All experiments were carried out according to the guidelines from Directive 2010/63/EU of the European Parliament on the protection of animals used for scientific purposes and approved by the local authorities for animal welfare (permission numbers for tissue and organ harvesting: twz 07-2017, twz 31-2017 and twz 13-2020). For breeding, animals were maintained at 12-h light and dark cycles with free access to food and water.

Neonatal primary microglial cells were isolated from the cerebral cortex of newborn mice (6–12 newborn male and female mice brains were pooled, respectively) as described previously [10,34]. Briefly, newborn mice were decapitated, and heads were transferred into Petri dishes filled with ice-cold phosphate buffered saline (PBS). Using fine scissors, the scalp was opened carefully along the midline, and the brain was removed. Then, meninges were removed, and cortices and hippocampi were collected in 15 mL tubes filled with PBS. Collected brains were processed in 2 mL dissociation media containing 200 μL 2.5% trypsin and further supplemented with 20 μL of DNAse I in order to digest DNA released from dead cells. Mixed cultures from newborn mouse brains were prepared and microglial cells were enriched as described [58,59]. After incubation at 37 °C and 5% CO_2_ for 30 min, the medium was removed, and the brain tissues were suspended in 2 mL of the Dulbecco’s modified Eagle’s medium (DMEM, Sigma-Aldrich, #06429, endotoxin-tested) containing 10% heat-inactivated fetal bovine serum (FBS, Sigma-Aldrich, #F7524, endotoxin-free and sterile-filtered), 1% penicillin/streptomycin, 1% amphotericin B, supplemented with 30 μL DNAse I. Brain tissues were then homogenized and further transferred to T75 cell culture flasks with additional 8 mL of a culture medium and incubated at 37 °C and 5% CO_2_ for seven days, followed by medium change and further incubation for seven more days. Then, adherent microglial cells were separated from astrocytes by adding a PBS–EDTA solution and careful shaking. After harvesting, microglial cells were seeded in adherent well plates. Purity of microglia was in the range between 94 and 98%, as confirmed by Iba1 staining (Appendix A).

### 4.2. Microglial Cell Stimulation

Microglial cells (75 000 cells/well) were seeded in a 12-well plate and treated according to the stimulation scheme depicted in Figure 1A [9,46]. The cells were stimulated twice following a two-step (“two-hit”) protocol. Microglia were initially stimulated (“primed”) with different doses of LPS (“first hit”; 1 fg/mL or 100 ng/mL for 24 h, respectively; *Escherichia coli* serotype 055:B5 obtained from Sigma-Aldrich, St. Louis, MO, USA). The cells were restimulated five days after the first challenge (day 6) by a fixed dose of LPS (“second hit”; 100 ng/mL). Microglial cells were divided into four groups: microglia of the first group was used unstimulated (US group); the second group included unprimed microglia (UP group, without the “first hit” on day 1, but stimulated on day 6 with a fixed dose of LPS, 100 ng/mL); the third group (ULP group) included microglia stimulated with an ultra-low (1 fg/mL LPS) dose of stressors on day 1 and restimulated on day 6 with a fixed dose of LPS (100 ng/mL); the fourth group labeled high-dose-primed group (HP group) was stimulated with a high dose (100 ng/mL LPS) of stressors and restimulated on day 6 with a fixed dose of LPS (100 ng/mL). 

### 4.3. Antibodies

Monoclonal antibodies against the catalytic subunit p110γ of PI3Kγ was produced in our laboratory [23]. Primary antibodies for phospho-CREB (#9198) and CREB (#9104) were purchased from Cell Signaling (Danvers, MA, USA). Goat polyclonal anti-Iba-1 antibody (#ab5076, Abcam, Cambridge, UK) was used for Iba1 staining. Antibodies against β-actin (#A2228 and #A5441) were obtained from Sigma-Aldrich (St. Louis, MO, USA). Secondary HRP-coupled anti-rabbit and anti-mouse antibodies were purchased from KPL (Weden, Germany).

### 4.4. SDS-PAGE Western Blotting

The cells were lysed using a RIPA buffer containing 50 mM Tris/HCl, pH 8; 150 mM NaCl, 1% (*v*/*v*) NP-40, 0,5% (*v*/*v*) Na-deoxycholate, 0,1% (*w*/*v*) SDS, 100 mg/mL Pefa-Block, 1 mg/mL pepstatin A, 10 mM sodium vanadate and 1 mg/mL leupeptin. Samples were centrifuged (13,500× *g* for 30 min at 4 °C), and supernatants were mixed with 5× protein sample buffer (5% SDS, 33% glycerol, 25% β-mercaptoethanol, 83 mM Tris-HCl with pH up to 6.8 and 0.1 mg/mL bromophenol blue) and heated for 5 min at 95 °C. Protein samples were separated on 10 % polyacrylamide gel, transferred to a 0.45 μm polyvinylidene fluoride (PVDF) membrane and then immunoblotted with the abovementioned primary antibodies. Protein bands were detected by enhanced chemiluminescence reaction using a LAS4000 camera (Fuji Photo Film Co., Tokyo, Japan). Quantification of the protein bands on the membrane was completed using the Fujifilm Multi Gauge ver. 3.0 software (Fuji Photo Film Co., Tokyo, Japan).

### 4.5. Seahorse Assay and Crystal Violet Staining

Metabolic flux analysis was performed using a Seahorse XF96 Analyzer (Agilent Technologies Germany GmbH & Co. KG, Waldbronn, Germany). 20,000 primary microglial cells per well were seeded on 96-well Seahorse XF96 cell culture microplates. The cells were prepared and treated as described above (4.2 microglial cell stimulation). Before measurement, the supernatant medium was replaced with the Seahorse XF Assay Medium (pH adjusted to 7.4, supplemented with 10 mM D-glucose and 1 mM sodium pyruvate). The cells were then incubated for one additional hour in a CO_2_-free incubator at 37 °C. The oxygen consumption rate (OCR) and extracellular acidification rate (ECAR) were monitored at basal conditions and after sequential administration of oligomycin (2 μM; Abcam, Cambridge, UK) in order to block the mitochondrial ATP synthase. Thereafter, 2,4-dinitrophenol (30 μM; 2,4-DNP; Sigma-Aldrich Chemie GmbH, Taufkirchen, Germany) was added to uncouple mitochondrial respiration from ATP production; subsequently, antimycin A (2 μM; Sigma-Aldrich Chemie GmbH, Taufkirchen, Germany) was added to fully inhibit mitochondrial respiration. The inhibitors were injected directly after the last measurement data point of each cycle (basal conditions; oligomycin; 2,4-DNP; antimycin A). Timepoints were as follows: basal conditions: 1.35 min, 8.12 min, 14.85 min; oligomycin: 21.80 min, 28.53 min, 35.27 min; 2,4-DNP: 42.22 min, 48.95 min, 55.67 min; antimycin A: 62.63 min, 69.35 min, 76.08 min. The following parameters were obtained: (i) mitochondrial ATP-linked respiration, which is defined as the estimation rate of oxygen consumption to drive mitochondrial ATP synthesis (represents the difference of basal and proton leak values; specifically, the difference between OCR values under basal conditions and after mitochondrial ATP synthase blockade by oligomycin); (ii) the non-mitochondrial respiration expresses oxygen consumption rates after the blockage of mitochondrial respiration upon addition of complex III of the electron transport chain inhibitor such as antimycin A (represents OCR values after antimycin A administration); (iii) proton leak displays the “leak” of protons back across the inner membrane of mitochondria, which stimulates the activity of the respiratory chain independently of ATP synthetase (represents the difference between oligomycin and antimycin A values of the OCR). The spare (reserve) capacity is defined as the difference between basal and maximal respiration indicating the ability of the cell to meet increased energy demand (represents the difference between 2,4-dinitrophenol and basal OCR values). The maximal respiration displays the maximum rate of respiration responding thus to an increased energy demand (represents the difference between 2,4-dinitrophenol and antimycin A values of the OCR). Glycolytic reserve capacity indicates the capability of a cell regarding the energy demand (represents the difference between 2,4-dinitrophenol and basal ECAR values).

Measurements were performed for 1.5 h in 3 min mix and 3 min measure cycles at 37 °C in ten biological replicates per condition. At the end of the measurement, cell densities per well were quantified by crystal violet staining. Therefore, fixation of cells was performed using 100% ice-cold methanol at room temperature (RT) for 10 min. The cells were incubated for 30 min at 37 °C in a 0.05% crystal violet solution (in H_2_O; Sigma-Aldrich) and washed two times with H_2_O. Subsequently, cell-bound crystal violet was resolved in 10% acetic acid whilst being shaken on an orbital shaker. The resulting absorption was measured on an Infinite^®^ M1000 PRO microplate reader (Tecan, Männedorf, Switzerland). The observed OCR and ECAR were depicted as pmole/min and mpH/min, respectively, and normalized to the corresponding cell densities. The Wave software from Agilent Technologies was used to analyze the datasets. 

### 4.6. Measurement of Fatty Acid Oxidation (β-Oxidation)

Fatty acid oxidation in primary microglial cells was determined by a method described previously [60,61,62]. Briefly, 75,000 cells/well were seeded in a 24-well plate and stimulated according to the described stimulation protocol (see above). Twenty-four hours after the second stimulation, the medium was changed to 0.2 mL DMEM mixture (containing 0.25% free fatty acid-free bovine serum albumin (BSA), 50 μM L-carnitine and 500 pM biotin). Positive controls contained, in addition, 20 mM 2-deoxyglucose.

After five hours, the reaction was started by the addition of 20 μL [1-14C]-palmitate complexed to free fatty acid-free BSA in the presence of unlabeled palmitate (final concentration: 1.9 μCi/mL, 218 μM). The wells were then immediately covered with 4.41 cm^2^ Whatman paper grade 3 and incubated for two hours at 37 °C (5% CO_2_). Afterwards, the Whatman paper was moistened with 200 μL 3 M NaOH to capture the ^14^CO_2_ produced by β-oxidation, which was released from the cells by addition of 30 μL 70% perchloric acid per well. After overnight incubation at 4 °C, the Whatman paper was carefully placed in scintillation vials followed by the addition of 2 mL Rotiszint^®^ eco plus scintillation fluid and the radioactivity was analyzed using a Wallac 1414 WinSpectral Liquid Scintillation Counter (Jiaxing, China). Positive (paper containing 2 μL ^14^C-palmitate complex solution) and negative (not treated paper) controls were used for internal supervision/evaluation of the assay (data not shown). Protein content of cells was determined using a Pierce™ 660 nm Protein Assay Kit (see below) in lysates from parallel samples treated in the same experimental way. The amount of oxidized palmitate was calculated from the measured ^14^CO_2_ radioactivity normalized to protein amount and expressed as pmol palmitate/μg protein. 

### 4.7. RNA Isolation and Real-Time qPCR

To determine gene expression levels, total RNA was extracted six hours after the second stimulation with LPS (100 ng/mL) using a QIAzol Lysis Reagent (#79306) purchased from Qiagen (Hilden, Germany) following the manufacturer’s instructions. During the whole procedure, an RNase Away (#7003, Molecular BioProducts, San Diego, CA, USA) solution was used to flush pipettes and other equipment to prevent any contamination with other RNases or DNAs. RNA concentration and quality were checked by using the Nanodrop ND-1000 machine (Peqlab, Erlangen, Germany). Complementary DNA (cDNA) was synthesized using a RevertAid First Strand cDNA Synthesis kit (#K1612) from Thermo Fisher Scientific (Waltham, MA, USA). Real-time qPCR reaction was performed by using a StepOnePlus^TM^ real-time PCR System (Applied Biosystems, Foster City, CA, USA). Primers used in the study are listed in Table 1.

GAPDH was used as the housekeeping gene. Relative gene expression was calculated using the comparative C_T_ (2^−∆∆C^_T_) method [63].

### 4.8. Measurement of the Protein Concentration 

Total protein concentration was determined using the Pierce™ 660 nm Protein Assay Kit (#22662) from Thermo Fisher Scientific (Waltham, MA, USA). An ionic detergent compatibility reagent (IDCR) (#22663, Thermo Fischer Scientific, Waltham, MA, USA) was used in order to increase detergent compatibility and reduce interference. Briefly, 10 μL of the standard, sample and blank in duplicates were plated in a 96-well plate, followed by immediate addition of 150 μL assay reagent supplemented with IDCR. Then, the plate was covered and shaken for one minute in a plate shaker. Afterwards, the plate was incubated for additional five minutes at room temperature without shaking. Absorbance was measured at 660 nm using a TECAN Infinite 200 Plate reader (Tecan, Männedorf, Switzerland). Protein concentration is then calculated based on the values of the standard curve.

### 4.9. cAMP Measurements

Primary microglial cells (5000 cells/well) were seeded in 96-well plates in triplicates and stimulated as described above. Twenty-four hours after the second stimulation, cyclic AMP production was measured using a cAMP GloAssayKit (#V1501, Promega GmbH, Walldorf, Germany) following the manufacturer’s protocol. Measurement of luminescence was performed using a TECAN Infinite 200 Plate reader (Tecan, Männedorf, Switzerland), and the cAMP values were calculated based on the values of the standard curve.

### 4.10. Lactate Production Measurements

Supernatants from the microglial culture were used to measure lactate production by sequential enzymatic reactions as described in [64]. Briefly, first, the lactate was converted by lactate oxidase (LO; #L0638, Merck, Darmstadt, Germany) to pyruvate and H_2_O_2_. In the second reaction, the chromogenic substrate 2,2′-azino-di-(3-ethylbenzthiazoline sulfonic acid) (ABTS) (#A1888, Merck, Darmstadt, Germany) was converted to a colored dye, catalyzed by horseradish peroxidase (HRP; #77332, Merck, Darmstadt, Germany) in presence of H_2_O_2_ and measured at 405 nm. Lactate concentration was then calculated from a standard curve of known concentrations of lactate (Lactate Standard for IC TraceCERT; #07096, Merck, Darmstadt, Germany).

### 4.11. Analysis of Cell Viability by MTT Assay

Cell viability was determined using the MTT assay. The cells were seeded into a 96-well plate and incubated at 37 °C (5% CO_2_) for 24 h. After attachment, primary microglial cells were treated according to the stimulation scheme depicted in Figure 1A. Twenty-four hours after the second LPS stimulation, an MTT (3-(4,5-dimethylthiazol-2-yl)-2,5-diphenyltetrazolium bromide) solution was added and incubated for four hours at 37 °C (5% CO_2_). Next, a solubilization solution was added to each well and incubated overnight at 37 °C (5% CO_2_) for 24 h. Absorbance was measured at 570 nm using a TECAN Infinite 200 Plate Reader (Tecan, Switzerland) and data were shown as relative viability (wild-type UP assigned as 100%) (Appendix A).

### 4.12. Statistical Analysis

Statistical analysis was carried out using SigmaPlot Software version 13.0, build 13.0.0.83 (Systat Software GmbH, Erkrath, Germany). If not otherwise noted, data are presented as means ± SD. The Grubbs’s test was used to remove extreme outliers. Differences between the experimental groups and conditions were tested with a one-way or two-way analysis of variance, accordingly. Post-hoc comparisons were made with the Holm–Sidak test. Differences were considered significant with *p* < 0.05.

## 5. Conclusions

The aim of this study was to elaborate on and highlight specifically components of immunometabolic reprogramming of microglia for the induction of IIM. Our data reveal that LPS-induced IIM was accompanied by a marked OCR enhancement and increased ATP production due to ULP. In contrast, HP was followed by suppressed OCR and glycolytic activity indicative of immune tolerance. Furthermore, aerobic glycolysis and β-oxidation support opposing energy turnover by LPS-induced IIM. Intriguingly, PI3Kγ provoked a marked inhibition of glycolysis due to modulation of cAMP-dependent pathways. However, we did not observe any significant effect of PI3Kγ signaling on the LPS dose-dependent priming of immunometabolic rewiring.

## Figures and Tables

**Figure 1 ijms-22-02578-f001:**
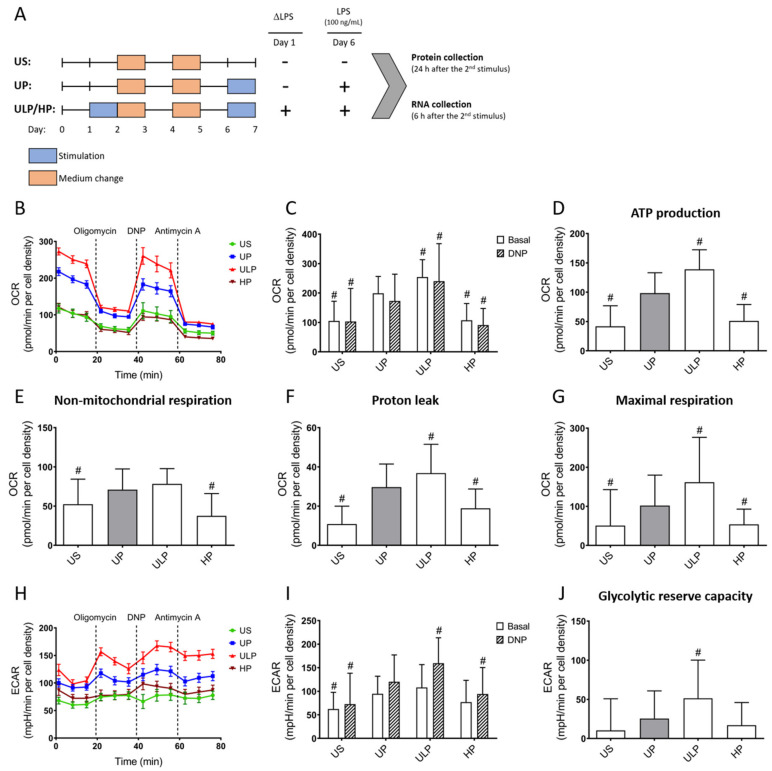
Reversed metabolism as a hallmark of LPS-induced innate immune memory (IIM). (**A**) Illustration of the two-step approach to induce memory-like inflammatory responses in primary microglial cell cultures. Measurement of (**B**) oxygen consumption rates (OCR) with (**C**) quantification under basal and 2,4-dinitrophenol (DNP, inducing uncoupled mitochondrial respiration) treatment. (**H**) Extracellular acidification rates (ECAR) with (**I**) quantification under basal and DNP treatment (n = 6, repeated measurements were performed 24 h after the second stimulus with LPS (100 ng/mL) using a Seahorse assay). (**D**) Mitochondrial ATP production, (**E**) non-mitochondrial respiration, (**F**) proton leak and (**G**) maximal respiration were calculated from OCR values. (**J**) Glycolytic reserve capacity was calculated from respective ECAR values. Microglia isolated from newborn mice were primed initially by ultra-low (ULP, 1 fg/mL) or high (HP, 100 ng/mL) doses of LPS, followed by the second stimulation (day 6) with 100 ng/mL LPS. Data were normalized and compared to unprimed microglia (UP group without any stimulation at day 1 with stimulation at day 6 with a fixed dose of LPS, 100 ng/mL). Unstimulated microglia (US) served as the negative control. Data from B, H are given as means ± SEM, data from (**C**–**G**,**I**,**J**) are presented as means + SD, ^#^
*p* < 0.05 vs. unprimed conditions (UP, grey column).

**Figure 2 ijms-22-02578-f002:**
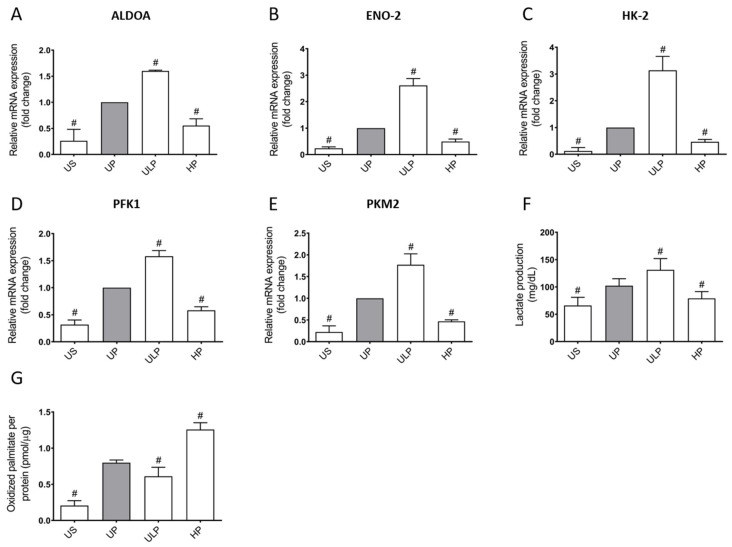
Aerobic glycolysis and β-oxidation of fatty acids support opposing memory-like inflammatory responses in LPS-primed microglia. Primary microglial cells were analyzed for gene expression of (**A**) aldolase A (ALDOA), (**B**) enolase 2 (ENO-2), (**C**) hexokinase 2 (HK-2), (**D**) phosphofructokinase 1 (PFK1), (**E**) pyruvate kinase M2 (PKM2) (n = 4; real-time PCR, normalized to glyceraldehyde-3-phosphate dehydrogenase (GAPDH) at unprimed state). (**F**) Lactate production (n = 7) and (**G**) fatty acid oxidation were measured (n = 3; normalized to protein concentration). Microglia isolated from newborn mice were primed initially by ultra-low (ULP, 1 fg/mL) or high (HP, 100 ng/mL) doses of LPS, followed by a second stimulation (day 6) with 100 ng/mL LPS. Data were compared to unprimed microglia (UP group without any stimulation at day 1 with stimulation at day 6 with a fixed dose of LPS, 100 ng/mL). Unstimulated microglia (US) served as the negative control. Data are presented as means + SD, ^#^
*p* < 0.05 vs. unprimed conditions (UP, grey column).

**Figure 3 ijms-22-02578-f003:**
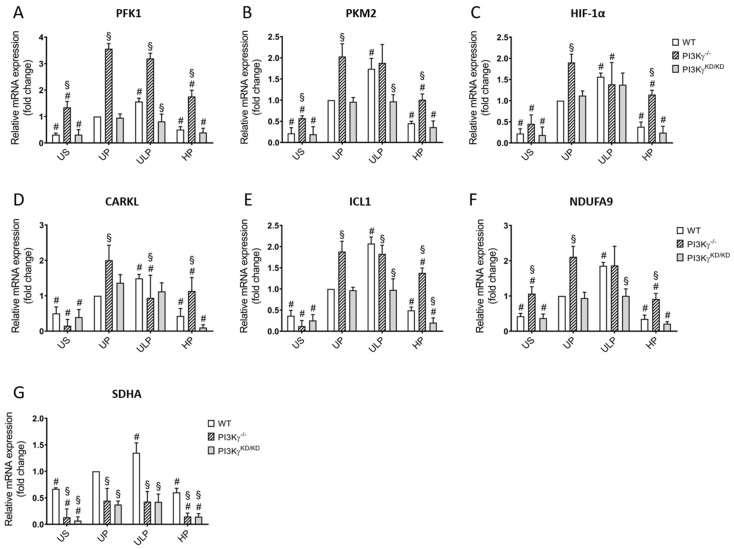
Lipid kinase-independent function of PI3Kγ mediates mainly priming-independent metabolic rewiring in microglial cells. HP priming induces immune tolerance. Primary microglial cells were analyzed for gene expression of (**A**) phosphofructokinase 1 (PFK1), (**B**) pyruvate kinase M2 (PKM2), (**C**) hypoxia factor 1-α (HIF-1α), (**D**) carbohydrate kinase-like (CARKL), (**E**) isocitrate lyase 1 (ICL1), (**F**) NADH: ubiquinone oxidoreductase, subunit A9 (NDUFA9), (**G**) succinate dehydrogenase complex, subunit A (SDHA) (real-time PCR, normalized to GAPDH at wild-type unprimed state). Microglia isolated from newborn mice (wild-type, open columns; PI3Kγ^-/-^, hatched columns; PI3Kγ^KD/KD^, dark gray columns; n = 4, respectively) were primed initially by ultra-low (ULP, 1 fg/mL) or high (HP, 100 ng/mL) doses of LPS, followed by the second stimulation (day 6) with 100 ng/mL LPS. Data were normalized and compared to unprimed microglia (UP group without stimulation at day 1 and with stimulation at day 6 with a fixed dose of LPS, 100 ng/mL). Unstimulated microglia (US) served as the negative control. Data are presented as means + SD, ^#§^
*p* < 0.05, ^#^ vs. unprimed condition, ^§^ vs. the wild-type strain.

**Figure 4 ijms-22-02578-f004:**
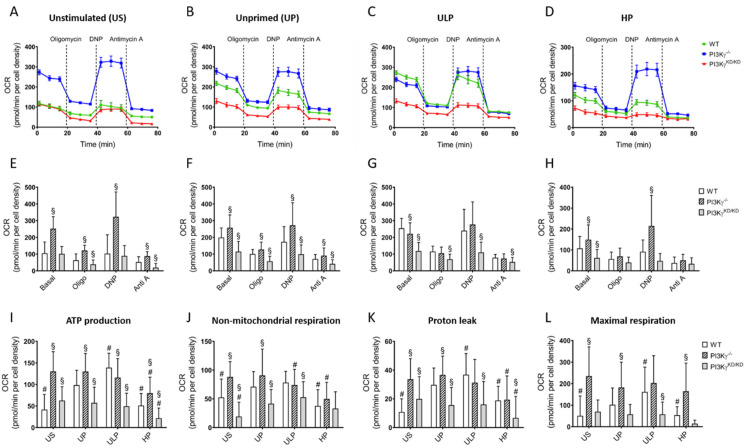
Lipid kinase-independent function of PI3Kγ mainly mediates control of microglial oxygen consumption (OCR). Microglia isolated from newborn mice (wild-type, open columns; PI3Kγ^-/-^, hatched columns; PI3Kγ^KD/KD^, dark gray columns; n = 5–6, repeated measurements, respectively) were primed initially by ultra-low (ULP, 1 fg/mL) or high (HP, 100 ng/mL) doses of LPS, followed by the second stimulation (day 6) with 100 ng/mL LPS. Time-dependent OCR (**A**–**D**) dynamics were assayed using a Seahorse assay and quantified under unstimulated (**E**), unprimed (**F**), ULP (**G**) and HP (**H**) conditions. (**I**) Mitochondrial ATP production, (**J**) non-mitochondrial respiration, (**K**) proton leak and (**L**) maximal respiration were calculated from OCR values. Unstimulated microglia (US) served as the negative control. Data from (**A**–**D**) are given as means ± SEM, data from (**E**–**L**) are represented as means + SD, ^#§^
*p* < 0.05, ^#^ vs. unprimed condition, ^§^ vs. the wild-type strain.

**Figure 5 ijms-22-02578-f005:**
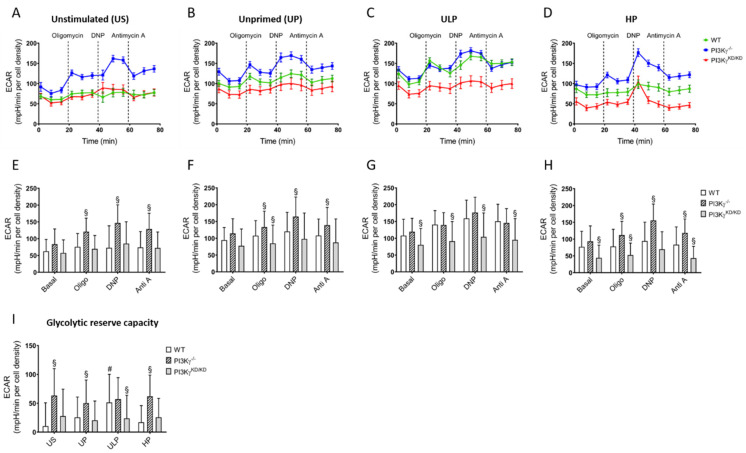
Lipid kinase-independent function of PI3Kγ mainly mediates control of microglial glycolytic activity (ECAR). Microglia isolated from newborn mice (wild-type, open columns; PI3Kγ^-/-^, hatched columns; PI3Kγ^KD/KD^, dark gray columns; n = 5–6, repeated measurements, respectively) were primed initially by ultra-low (ULP, 1 fg/mL) or high (HP, 100 ng/mL) doses of LPS, followed by the second stimulation (day 6) with 100 ng/mL LPS. Time-dependent ECAR (**A**–**D**) dynamics were assayed using a Seahorse assay and quantified under unstimulated (**E**), unprimed (**F**), ULP (**G**) and HP (**H**) conditions. (**I**) Glycolytic reserve capacity was calculated from the ECAR values. Unstimulated microglia (US) served as the negative control. Data from (**A**–**D**) are given as means ± SEM, data from (**E**–**I**) are represented as means + SD, ^#§^
*p* < 0.05, ^#^ vs. unprimed condition, ^§^ vs. the wild-type strain.

**Table 1 ijms-22-02578-t001:** Primers and their sequences used in this study.

Gene Name	Primer Sequences (5′–3′)	
***ALDOA*** **(Aldolase A)**	Forward:	CAACGGTCACAGCACTTCG
	Reverse:	GGCTCGACCATAGGAGAAAG
***CARKL***	Forward:	CAGGCCAAGGCTGTGAAT
(Carbohydrate kinase-like)	Reverse:	GCCAGCTGCATCATAGGACT
***ENO-2***	Forward:	TGGCAAGGATGCCACTAACGTG
(Enolase 2)	Reverse:	AACTCAGAGGCAGCCACATCCA
***GAPDH***	Forward:	CATGGCCTTCCGTGTTTCCTA
(Glyceraldehyde-3-phosphate dehydrogenase)	Reverse:	CCTGCTTCACCACCTTCTTGAT
***HIF-1α***	Forward:	CTCATCAGTTGCCACTTCC
(Hypoxia factor -1 α)	Reverse:	TCATCTTCACTGTCTAGACCAC
***HK-2*** (Hexokinase 2)	Forward:	ATTGTCCAGTGCATCGCGGA
	Reverse:	AGGTCAAACTCCTCTCGCCG
***ICL1***	Forward:	ACCCAGCCTTTGGATGAAGG
(Isocitrate lyase 1)	Reverse:	GTTACAGAGGTGGGACGCAA
***NDUFA9***	Forward:	TCCGCTTTCGGGTTGTTA
(NADH: Ubiquinone oxidoreductase, subunit A9)	Reverse:	GTACCGGTTTGGCCCAGT
***PFK1***	Forward:	TGACATGACCATTGGCACAG
(Phosphofructokinase 1)	Reverse:	TCTTGCTACTCAGGATTCGG
***PKM2***	Forward:	GTCTGGAGAAACAGCCAAGG
(Pyruvate kinase M2)	Reverse:	CGGAGTTCCTCGAATAGCTG
***SDHA***	Forward:	AACACTGGAGGAAGCACACC
(Succinate dehydrogenase complex, subunit A)	Reverse:	AGTAGGAGCGGATAGCAGGA

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
