# Peer review of "The Role of the Pathogen Dose and PI3Kγ in Immunometabolic Reprogramming of Microglia for Innate Immune Memory"

_ijms, 2021, doi:10.3390/ijms22052578_

Round 1

Reviewer 1 Report

This is a very interesting work which definitely merits publication. 

However, there are several points which need to be addressed.

Please see my specific comments/suggestions below on how to improve the work.  

1) In the introduction the Authors present microglia as innate immune cells. It is correct to say that they play crucial role in innate immune responses, but this is not the only function of these cells. They are also known as key cells playing a role in overall brain maintenance. Microglial cells are constantly scavenging the CNS for damaged or unnecessary cells (neurons), synapses, plaques as well as infectious agents. This needs to be reflected/stated not to confuse the readers who might be interested in the work but don't know much about microglia - the Journal is aimed at wide audience. 

2) In the introduction please also briefly explain the role of other PI3K isoforms and explain why you are focusing on the gamma one and not the other isoforms. 

3) First section of the Results and all the way through the section - where is the evidence that LPS induces innate immune memory? It is not clear from the results. Please explain the statement or re-word it saying "innate immune response" instead of "...memory".

4) Figure 2 title. "Aerobic glycolysis and beta-oxidation..." Please be specific - and say beta oxidation of fatty acids, otherwise it is not clear what is beta-oxidised. 

5) Section 2.2 and Figure 2. The conclusions on the activity of aerobic glycolysis in LPS – induced IIM are drawn from mRNA levels of the enzymes. Biochemically this is incorrect. Activities of respective enzymes (at least some) need to be measured. What the Authors have done - they just measured lactate production. Lactate is the product of anaerobic not aerobic glycolysis. 

6) The same concerns regarding the Figure 3. Please explain why you think measuring mRNA is enough. For example HIF-1alpha: - no matter how high are the levels of its mRNA, translated protein gets degraded in the presence of oxygen with the help of a group of proteins including HIF-1alpha prolyl-hydroxylases and von Hippel-Lindau protein which lead the protein to proteasomal degradation. And only inhibition of these processes (for example by hypoxia) leads to HIF-1alpha stabilisation/accumulation.

7) In the conclusions please explain why you always refer to LPS-induced innate immune memory and not innate immune responses. 

Author Response

#Reviewer 1:

This is a very interesting work which definitely merits publication. 

However, there are several points which need to be addressed.

Please see my specific comments/suggestions below on how to improve the work.

1) In the introduction the Authors present microglia as innate immune cells. It is correct to say that they play crucial role in innate immune responses, but this is not the only function of these cells. They are also known as key cells playing a role in overall brain maintenance. Microglial cells are constantly scavenging the CNS for damaged or unnecessary cells (neurons), synapses, plaques as well as infectious agents. This needs to be reflected/stated not to confuse the readers who might be interested in the work but don't know much about microglia - the Journal is aimed at wide audience.

Response: We thank the reviewer for the valuable recommendations. The mentioned issues are now thoroughly considered in the revised manuscript as suggested (please refer to lines 41-44).

2) In the introduction please also briefly explain the role of other PI3K isoforms and explain why you are focusing on the gamma one and not the other isoforms.

Response: We thank the reviewer for the valuable recommendations. The mentioned issues are now thoroughly considered in the revised manuscript as suggested (please refer to lines 73-87 & 89-92 & 98-103).

3) First section of the Results and all the way through the section - where is the evidence that LPS induces innate immune memory? It is not clear from the results. Please explain the statement or re-word it saying "innate immune response" instead of "...memory".

Response: The chosen experimental design using approaches of initial priming with ultra-low and high doses of LPS (ULP or HP, respectively) on day 1 and re-stimulation at day 6 with a fixed dose of LPS (also called “two-hit” protocol [1, 2]) implies innate immune memory, if relevant differences in immune response occur between primed versus unprimed conditions. Accordingly, our measuring procedure warranted a five days persistent modification of microglial immune response – therefore manifestation of induced innate immune memory [3]. In order to clarify this issue, we added respective comments in chapter “Introduction” (please refer to lines 47-53, 58-59).

4) Figure 2 title. "Aerobic glycolysis and beta-oxidation..." Please be specific - and say beta oxidation of fatty acids, otherwise it is not clear what is beta-oxidised.

Response: We thank the reviewer for this information. The omission has been addressed.

5) Section 2.2 and Figure 2. The conclusions on the activity of aerobic glycolysis in LPS – induced IIM are drawn from mRNA levels of the enzymes. Biochemically this is incorrect. Activities of respective enzymes (at least some) need to be measured. What the Authors have done - they just measured lactate production. Lactate is the product of anaerobic not aerobic glycolysis.

Response: We agree with the reviewer that it is inconclusive to argue that gene expression data / mRNA levels of key glycolytic enzymes are indicative for specific activity levels of aerobic glycolysis in LPS – induced IIM. However, this is not what we did. We just refer to similarities in expression patterns of key (partly rate-controlling) glycolytic enzymes and microglial ECAR responses.

We respectfully disagree with the Reviewer’s point that measurement of lactate production reflected solely the anaerobic glycolysis in our experiments:

First, microglial activation occurs usually in vivo under aerobic conditions and always during performance of the Seahorse analysis and cell culture measurements in this study [4, 5] There is compelling evidence that microglial activation induces a marked increase of the glycolytic metabolism. Indeed, glycolysis is one of the major metabolic pathways in all cell types including microglia provide energy and building blocks for essential biosynthetic pathways [6]. Although glycolysis is much less efficient in ATP production than OXPHOS, it is the major pathway for energy production in proliferating cells, e.g. tumor cells (Warburg effect). The same glycolytic preference holds true for proinflammatory activated immune cells [7]. A prerequisite for glycolytic energy production is that glycolysis can be rapidly activated via the induction of enzymes that are involved in this pathway (in contrast to OXPHOS where new mitochondria has to be produced, which needs much more time and resources).

Increased lactate production occurs during aerobic glycolysis due to proinflammatory immune cell activation in order to maintain glycolytic flux due to pyruvate reduction to lactate to recycle NADH and maintain NAD+ levels. This is used by numerous enzymes as a cofactor, as well as enabling the diversion of intermediate products to biosynthetic growth pathways to support anabolic growth [5]. Therefore, enhanced glycolysis enables the immune cell to generate sufficient ATP and biosynthetic intermediates to carry out its particular effector functions.

Conversion of pyruvate to lactate takes place because metabolic reprogramming of proinflammatory activated immune cells involves blockade of the TCA cycle by inhibition of the isocitrate lyase leading to citrate accumulation feeding production of fatty acids and succinate accumulation [8, 9]. Therefore, maintenance of a high glycolytic flux needs lactate production, which is representative for glycolytic activity under aerobic conditions.

6) The same concerns regarding the Figure 3. Please explain why you think measuring mRNA is enough. For example HIF-1alpha: - no matter how high are the levels of its mRNA, translated protein gets degraded in the presence of oxygen with the help of a group of proteins including HIF-1alpha prolyl-hydroxylases and von Hippel-Lindau protein which lead the protein to proteasomal degradation. And only inhibition of these processes (for example by hypoxia) leads to HIF-1alpha stabilisation/accumulation.

Response: The objective of this manuscript was to illustrate the quite complex patterns of metabolic reprogramming in microglia for innate immune memory depending on priming dose and PI3Kγ signaling. This includes, as the main components, the glycolytic metabolic pathway, the TCA cycle and the electron transport chain, the pentose phosphate pathway and fatty acid oxidation, which we addressed accordingly.

We decided to focus mainly on early components of metabolic reprogramming to be able to detect a rather broad spectrum of possibly fine-tuned regulatory effects using sensitive measures.

We are aware that a gap in knowledge remains regarding the relevance of these findings for the microglial function. This gap will be addressed in future studies. Nevertheless, presented data clearly show that PI3Kγ is involved in regulation of different parts of energy metabolism. However, we detected no impact of PI3Kγ signaling on immunometabolic rewiring due to dose-dependent LPS priming. We addressed this aspect in the “Discussion” chapter (please refer to lines 295-298).

7) In the conclusions please explain why you always refer to LPS-induced innate immune memory and not innate immune responses.

Response: We refer to LPS-induced innate immune memory because the objective of this study was to elaborate and highlight specifically this aspect of microglial immune response.

This was outlined in the “Conclusion” chapter (please refer to lines 551-552).

References

  1. Lajqi, T.; Lang, G. P.; Haas, F.; Williams, D. L.; Hudalla, H.; Bauer, M.; Groth, M.; Wetzker, R.; Bauer, R., Memory-Like Inflammatory Responses of Microglia to Rising Doses of LPS: Key Role of PI3Kgamma. Front Immunol 2019, 10, 2492.
  2. Schaafsma, W.; Zhang, X.; van Zomeren, K. C.; Jacobs, S.; Georgieva, P. B.; Wolf, S. A.; Kettenmann, H.; Janova, H.; Saiepour, N.; Hanisch, U. K.; Meerlo, P.; van den Elsen, P. J.; Brouwer, N.; Boddeke, H. W.; Eggen, B. J., Long-lasting pro-inflammatory suppression of microglia by LPS-preconditioning is mediated by RelB-dependent epigenetic silencing. Brain, behavior, and immunity 2015, 48, 205-21.
  3. Neher, J. J.; Cunningham, C., Priming Microglia for Innate Immune Memory in the Brain. Trends in immunology 2019, 40, (4), 358-374.
  4. Divakaruni, A. S.; Paradyse, A.; Ferrick, D. A.; Murphy, A. N.; Jastroch, M., Analysis and interpretation of microplate-based oxygen consumption and pH data. Methods in enzymology 2014, 547, 309-54.
  5. O'Neill, L. A.; Kishton, R. J.; Rathmell, J., A guide to immunometabolism for immunologists. Nature reviews. Immunology 2016, 16, (9), 553-65.
  6. Riksen, N. P.; Netea, M. G., Immunometabolic control of trained immunity. Molecular aspects of medicine 2020, 100897.
  7. Cheng, S. C.; Quintin, J.; Cramer, R. A.; Shepardson, K. M.; Saeed, S.; Kumar, V.; Giamarellos-Bourboulis, E. J.; Martens, J. H.; Rao, N. A.; Aghajanirefah, A.; Manjeri, G. R.; Li, Y.; Ifrim, D. C.; Arts, R. J.; van der Veer, B. M.; Deen, P. M.; Logie, C.; O'Neill, L. A.; Willems, P.; van de Veerdonk, F. L.; van der Meer, J. W.; Ng, A.; Joosten, L. A.; Wijmenga, C.; Stunnenberg, H. G.; Xavier, R. J.; Netea, M. G., mTOR- and HIF-1alpha-mediated aerobic glycolysis as metabolic basis for trained immunity. Science 2014, 345, (6204), 1250684.
  8. Jha, A. K.; Huang, S. C.; Sergushichev, A.; Lampropoulou, V.; Ivanova, Y.; Loginicheva, E.; Chmielewski, K.; Stewart, K. M.; Ashall, J.; Everts, B.; Pearce, E. J.; Driggers, E. M.; Artyomov, M. N., Network integration of parallel metabolic and transcriptional data reveals metabolic modules that regulate macrophage polarization. Immunity 2015, 42, (3), 419-30.
  9. Tannahill, G. M.; Curtis, A. M.; Adamik, J.; Palsson-McDermott, E. M.; McGettrick, A. F.; Goel, G.; Frezza, C.; Bernard, N. J.; Kelly, B.; Foley, N. H.; Zheng, L.; Gardet, A.; Tong, Z.; Jany, S. S.; Corr, S. C.; Haneklaus, M.; Caffrey, B. E.; Pierce, K.; Walmsley, S.; Beasley, F. C.; Cummins, E.; Nizet, V.; Whyte, M.; Taylor, C. T.; Lin, H.; Masters, S. L.; Gottlieb, E.; Kelly, V. P.; Clish, C.; Auron, P. E.; Xavier, R. J.; O'Neill, L. A., Succinate is an inflammatory signal that induces IL-1beta through HIF-1alpha. Nature 2013, 496, (7444), 238-42.
  10. Schmittgen, T. D.; Livak, K. J., Analyzing real-time PCR data by the comparative C(T) method. Nature protocols 2008, 3, (6), 1101-8.

Reviewer 2 Report

Major comments:

  1. The language of whole manuscript should be revised by native English speaker. It was difficult to be followed.
  2. The conclusion of Abstract is not clear.
  3. In the introduction, what is the metabolic function of PI3Kγ in immune cells?
  4. How about the cell viability during the whole culture process and different treatment?
  5. How the gene expression value was calculated? Since it is fold change, why the values of fist bar are below 0.5, what is that means?

Author Response

# Reviewer 2:

Major comments:

  1. The language of whole manuscript should be revised by native English speaker. It was difficult to be followed.

Response: We got support from a colleague (Dr. Martin. G. Frasch, University of Washington: Seattle, US) who is expert for neuroimmunology and microglial cell function in brain inflammation and hypoxia. He revised and edited language issues.

  1. The conclusion of Abstract is not clear.

Response: We reworded the conclusion of the abstract (please refer to lines 31-33).

  1. In the introduction, what is the metabolic function of PI3Kγ in immune cells?

Response: We considered this issue thoroughly in response to the first reviewer’s 2nd concern (please refer to lines 73-87 & 89-92 & 98-103).

  1. How about the cell viability during the whole culture process and different treatment?

Response: We thank the reviewer for this notice of a forgotten information. We added it accordingly (please refer to lines 533-542 & Supplementary figure 3).

  1. How the gene expression value was calculated? Since it is fold change, why the values of fist bar are below 0.5, what is that means?

Response: The relative gene expression was calculated using the comparative CT (2−ΔΔCT) method as described [10]. The CT mean values of targeted genes have been normalized to the housekeeping gene GAPDH as suggested by Schmittgen et al. 2008 [10]. Then the unprimed (UP) state has been assigned as 1.0 and other data have been calculated related to UP.

Reviewer 3 Report

This research manuscript is interesting and well written.

Research plan is well designed and performed.

Results are clearly reported.

Discussion is well readable. 

Author Response

# Reviewer 3:

This research manuscript is interesting and well written.

Research plan is well designed and performed.

Results are clearly reported.

Discussion is well readable.

Response: Many thanks for the appreciative assessment.

Round 2

Reviewer 1 Report

After looking through the revised manuscript and the Authors' responses I recommend to accept this paper for publication.

Author Response

Response: Many thanks for the appreciative assessment.

Reviewer 2 Report

  1. In the seahorse assay of Figure 1, it is not clear what time add the drugs. How the respiration goes down before add oligomycin? Why the maximal respiration is almost the same value with basal level and the spare capacity is negative value after and DNP? This is not consistent with usual seahorse experiments.

  1. What is the point to measure the ECAR by adding DNP and Antimycin A, since they are just involved in mitochondria respiration, instead of glycolysis ?

Author Response

# Reviewer 2:

Comments and Suggestions for Authors:

No 1:

  • In the seahorse assay of Figure 1, it is not clear what time add the drugs.

Response: The inhibitors were injected directly after the last measurement data point of each cycle (basal conditions; oligomycin; 2,4-DNP; antimycin A). Time points were: basal conditions -1,35 min, 8,12 min, 14,85 min; oligomycin - 21,80 min, 28,53 min, 35,27 min; 2,4-DNP - 42,22 min, 48,95 min, 55,67 min; antimycin A - 62,63 min, 69,35 min, 76,08 min. We added the information to the re-revised manuscript (please refer to lines 442-446)

  • How the respiration goes down before add oligomycin?

Response: This happens in all three cycles that the first data point is higher and the values decrease during the cycle. This may be microglia specific, since we observed it in several independent assays. We ruled out artefacts from background corrections, which may occur, in each case.

In previous contributions using primary microglial cells derived from mice as well as rats and using immortal microglial cell lines (BV2 and B6M7 cells) exhibited partly similar patterns of OCR traces during basal conditions [1-5].

  • Why the maximal respiration is almost the same value with basal level and the spare capacity is negative value after and DNP?

Response: We thank the reviewer for the valuable recommendation. Seminal methodological contributions suggest the need to interpret spare respiratory capacity within a cell-specific context. In case that maximal respiration OCR values are high or nearly equal with basal OCR values may indicate a proliferating state of cells under consideration [6]. Under these conditions the spare capacities may appear negative due to the calculations. Re-checking relevant literature, we found that data sets with a negative spare respiratory capacity should be omitted from analysis for data because physically, maximal respiration must be greater than or equal to basal respiration. [7].

Accordingly, we decide to eliminate Fig. 1G from the re-revised manuscript

No 2: What is the point to measure the ECAR by adding DNP and Antimycin A, since they are just involved in mitochondria respiration, instead of glycolysis?

Response: During the analysis both OCR and ECAR are measured in parallel. So, it is valid and worth to check both readouts. The reviewer is right that the official assay to analyze glycolysis would be the Glyco Stress test. Nevertheless, our aim was to study mitochondrial performances and thus the Mito Stress Test is the best assay.

In addition, this procedure delivers valid information about the glycolysis at basal conditions as well as under maximum metabolic stress (after oligo/DNP injections), which then gives the information about their glycolytic spare capacity during metabolic stress [6].

References

  1. He, K.; Liang, X.; Wei, T.; Liu, N.; Wang, Y.; Zou, L.; Bai, C.; Yao, Y.; Wu, T.; Kong, L.; Zhang, T.; Xue, Y.; Tang, M., A metabolomics study: CdTe/ZnS quantum dots induce polarization in mice microglia. Chemosphere 2020, 246, 125629.
  2. Sekar, P.; Huang, D. Y.; Hsieh, S. L.; Chang, S. F.; Lin, W. W., AMPK-dependent and independent actions of P2X7 in regulation of mitochondrial and lysosomal functions in microglia. Cell communication and signaling : CCS 2018, 16, (1), 83.
  3. Wang, L.; Pavlou, S.; Du, X.; Bhuckory, M.; Xu, H.; Chen, M., Glucose transporter 1 critically controls microglial activation through facilitating glycolysis. Mol Neurodegener 2019, 14, (1), 2.
  4. Jaber, S. M.; Bordt, E. A.; Bhatt, N. M.; Lewis, D. M.; Gerecht, S.; Fiskum, G.; Polster, B. M., Sex differences in the mitochondrial bioenergetics of astrocytes but not microglia at a physiologically relevant brain oxygen tension. Neurochemistry international 2018, 117, 82-90.
  5. Rubio-Araiz, A.; Finucane, O. M.; Keogh, S.; Lynch, M. A., Anti-TLR2 antibody triggers oxidative phosphorylation in microglia and increases phagocytosis of beta-amyloid. J Neuroinflammation 2018, 15, (1), 247.
  6. Divakaruni, A. S.; Paradyse, A.; Ferrick, D. A.; Murphy, A. N.; Jastroch, M., Analysis and interpretation of microplate-based oxygen consumption and pH data. Methods in enzymology 2014, 547, 309-54.
  7. Nicholas, D.; Proctor, E. A.; Raval, F. M.; Ip, B. C.; Habib, C.; Ritou, E.; Grammatopoulos, T. N.; Steenkamp, D.; Dooms, H.; Apovian, C. M.; Lauffenburger, D. A.; Nikolajczyk, B. S., Advances in the quantification of mitochondrial function in primary human immune cells through extracellular flux analysis. PloS one 2017, 12, (2), e0170975.

Round 3

Reviewer 2 Report

The whole design need be improved greatly.